# Ozone-Ultrafine Bubbles for Reducing Concentration of Citric Acid and Sodium Chloride for Trimmed Young Coconut Preservation

**Supat Pathomaim [1], Suwatchai Jarussophon [2], Siwaret Arikit [3] and Wachiraya Imsabai [1,*]**

1. Department of Horticulture, Faculty of Agriculture at Kamphaeng Saen, Kasetsart University, Kamphaeng Saen Campus, Nakhon Pathom 73140, Thailand
2. National Nanotechnology Center, National Science and Technology Development Agency, Thailand Science Park, Pathum Thani 12120, Thailand
3. Department of Agronomy, Faculty of Agriculture at Kamphaeng Saen, Kasetsart University, Kamphaeng Saen Campus, Nakhon Pathom 73140, Thailand
* Correspondence: wachiraya.i@ku.th

**Abstract:** Citric acid (CA) and sodium chloride (NaCl) are used in organically trimmed young aromatic coconuts to prevent microbial growth and browning. However, the use of high concentrations of these chemicals is considered a waste and may elicit allergic reactions in the operator. This study aimed to reduce the concentration of these two substances by using a combination of ozone-ultrafine bubbles ($O_3$UFBs). The trimmed young coconuts were dipped in 20% CA + 20% NaCl (commercial method; $C_{20}N_{20}$), 20% CA + 10% NaCl + $O_3$UFBs ($C_{20}N_{10}$-$O_3$UFBs), and 15% CA + 15% NaCl + $O_3$UFBs ($C_{15}N_{15}$-$O_3$UFBs) for one minute. All the coconuts were wrapped with PVC film and stored at 2–4 °C for 30 days and then transferred to storage at 8–10 °C for 7 days. The quality of the coconut water and coconut meat was evaluated. The whiteness, browning index, polyphenol oxidase (PPO) activity, and total phenolic content of coconut mesocarp were investigated. Titratable acidity and the total soluble solid content of coconut water were 0.038–0.095% and 6.65–7.40 °Brix, respectively, while that of the coconut meat was 0.044–0.104% and 8.00–9.80 °Brix, respectively. The mesocarp whiteness, browning index, disease score, fruit appearance, PPO activity, and total phenolic content did not differ among the treatments. $C_{20}N_{10}$-$O_3$UFBs and $C_{15}N_{15}$-$O_3$UFBs treatments also controlled microbial growth and the surface browning of the trimmed coconuts. In conclusion, the use of $O_3$UFBs decreased the concentration of CA and NaCl by at least 25% of the commercial method used for treating trimmed young coconuts.

**Keywords:** organic; ultrafine bubble; browning; microbial; polyphenol oxidase

## 1. Introduction

The 'Nam Hom' coconut, or aromatic coconut (*Cocos nucifera* Linn.), is one of the most important crops exported from Thailand. The exported value in 2021 was around THB 9.3 billion [1]. 2-acetyl-1-pyrroline (2-AP), which is a key volatile component of the water and meat of coconut fruit, and smells similar to the pandan leaf and jasmine rice [2]. Three types of organic aromatic coconut are usually exported to other countries: green, trimmed, and polished coconuts. The green coconut has to be trimmed and cut into a diamond shape, or cylindrical shape for packing, storage, and transportation. Trimmed coconut is very popular due to its convenience for handling and transportation cost reduction [3]. Commercially trimmed organic coconuts are treated with 20% citric acid (CA) and 20% sodium chloride (NaCl) solutions to prevent microbial growth and surface browning during transportation and on-shelf in the market. However, high concentrations of these chemicals incur high costs and may cause allergies in the operator.

Recently, sulfite agents (e.g., sodium metabisulfite; SMS, potassium metabisulfite, and sulfur dioxide) and chlorine have been commonly used to extend the shelf-life of trimmed

coconut. However, these substances have been reported to cause allergenic effects [4,5]. Moreover, the USDA has banned the use of chemicals from the sulfite group on fresh-cut fruits [6]. In the past 5–10 years, many researchers have worked towards substituting SMS with better alternatives. A review of the literature shows that 2% oxalic acid with 0.2% benzoic can control browning and microbial growth on trimmed coconut when stored at 2 °C for 6 weeks. Similar effects can be seen when oxalic acid is combined with NaOCl and acetic acid. Nguyen et al. [7] reported that, besides SMS, treatment with 20% CA + 15% NaCl was the most effective treatment for maintaining visual quality and color, as well as controlling the microbial growth of trimmed coconut if stored at 2 °C for 8 weeks.

Ultrafine bubbles (UFBs) or nanobubbles (NBs) can be used to reduce the quantity of CA and NaCl when used for the abovementioned purpose. UFBs have a high surface area, which can enhance the chemicals' efficiency and remain suspended in water for a month. Saijai et al. [8] reported that fine bubbles of fresh ozone could control *Escherichia coli* in water. Ozone is already used in the drinking water industry to control microbial growth in water. Batagoda et al. [9] reported that nano-ozone bubbles decrease diffusion and increase ozone concentration in water. Ozone decomposition in water increases the oxygen ($O_2$) level and hydroxyl radicals that can oxidize in water. Yuk et al. [10] reported that ozone combined with organic acid successfully controlled *E. coli* and *Listeria monocytogenes* in lettuce stored at 15 °C for 10 days. Ozone is also a bleaching agent used in oxidizing agents or active oxygen compounds. Ultrafine bubbles of the ozone ($O_3$UFBs) might reduce the concentration of CA and NaCl to preserve trimmed young coconuts for longer.

Therefore, this research aims to prove the effectiveness of CA and NaCl combined with $O_3$UFBs in preserving trimmed aromatic coconuts when compared with the commercial method. Reducing the concentration of CA and NaCl (acid/salt solution) used for prolonging the shelf-life of trimmed young coconuts is beneficial for the environment and is cost-efficient.

## 2. Materials and Methods

### 2.1. Plant Material and Chemical Treatment

Young aromatic coconut fruits were harvested at 6–7 months after flowering. Then, the coconuts were delivered to a packing house located in Samut Sakhon Province, Thailand, where the fruits were trimmed with a sharp knife. They were cut at the top and shaped to form a cylindrical body and flat base, which is called the 'cylindrical' shape (Figure 1). The fruits weighed approximately 1.2–1.5 kg/fruit.

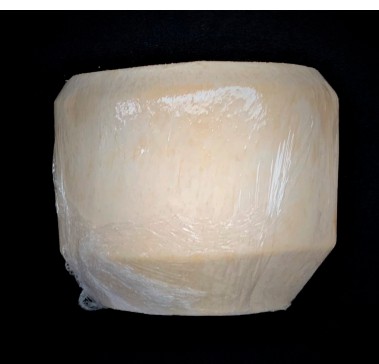

**Figure 1.** The green coconut fruit were trimmed as a 'cylindrical' shape.

Ozone-ultrafine bubble ($O_3$UFBs) water was generated using a corona discharge ozone generator with an ozone gas flow rate of 10 g/h, which was connected to an ultrafine bubble generator at a water flow rate of 50 L/min and operation pump pressure of 0.45 MPa. The $O_3$UFBs water generated from the cavitation nozzle was re-circulated in an $O_3$UFBs water tank. The concentration of dissolved ozone in the water was measured using an ozone test kit (Ozone Pack Test, Kyoritsu Rikagaku, Japan).

The citric acid (food grade), at concentrations of 20% and 15% (*w/v*), with NaCl at concentrations of 10% and 15% (*w/v*), were dissolved in ozone-ultrafine bubbles (O₃UFBs) and used in combination. After trimming, the coconuts were immediately dipped in the treatment solutions for 1 min: the lowest time for dipping in the processing line; the fruits were divided into three groups and dipped in 20% CA + 20% NaCl (commercial method; $C_{20}N_{20}$), 20% CA + 10% NaCl + O₃UFBs ($C_{20}N_{10}$-O₃UFBs), and 15% citric acid + 15% NaCl + O₃UFBs ($C_{15}N_{15}$-O₃UFBs). After dipping, the fruits were wrapped in an 11 μm thick PVC film and stored at $2 \pm 1\,°C$ and 80–85% RH, identical to the export conditions, for 30 days, and then transferred to 8–10 °C for 7 days (simulation of on-shelf conditions).

The fruits were investigated at 0, 3, and 7 days after transferring to 8–10 °C. The fruits from each treatment batch were checked in terms of the photo, disease scoring, mesocarp whiteness, and the quality of the coconut endosperm. The coconut mesocarp was sampled and frozen in liquid nitrogen and then stored at −70 °C until use. Each treatment batch had four biological replicates for fruit qualities by one fruit per replication.

### 2.2. Disease Score

The disease score was calculated based on the diseased area of mesocarp skin with the following scoring pattern: 1 = 0% of the diseased area (no disease), 2 = 1–10% of the diseased area, 3 = 11–20% of the diseased area, 4 = 21–30% of the diseased area, and 5 = ≥31% of the diseased area.

### 2.3. Mesocarp Whiteness (W) and Browning Index (BI)

The color changes (L *, a *, b *, and hue values) in the fruits from all treatment batches were determined by examining two spots in each fruit using a color meter (Minolta CR-400, Japan). The mesocarp whiteness was calculated using the following equation:

$$W = 0.511\,L* - 2.324\,a* - 1.100\,b* \qquad (1)$$

The Browning index (BI) [11] was calculated according to the following equation:

$$BI = [(X - 0.31)/0.172] \times 100 \qquad (2)$$

where

$$X = (a* + 1.75L*)/(5.646\,L* + a* - 3.012\,b*) \qquad (3)$$

### 2.4. Coconut Water and Coconut Meat Qualities

Liquid endosperm (coconut water) and solid endosperm (coconut meat) were used to record the total soluble solids (TSS), titratable acidity (TA), and turbidity during storage and after transferring to 10 °C.

The TSS of coconut water was measured using a hand refractometer (ATAGO, Japan). A total of 10 mL of coconut water was titrated with 0.1 N sodium hydroxide (NaOH), with 1% phenolphthalein as an indicator [12], and were expressed as the malic acid equivalent per ml. The turbidity of coconut water was measured as the percentage of transmittance (%T) using a spectrophotometer (Thermo Scientific, Waltham, MA, USA) at a 610 nm wavelength, as described by Campos et al. [13]. Reverse-osmosis water was used as a blank.

Solid endosperm (coconut meat) was used to record the TSS and TA. A total of 5 g of coconut meat were homogenized with 15 mL of reverse-osmosis (RO) water for 30 s and then centrifuged at 12,000 rpm 4 °C for 20 min. The supernatant was used for recording the TSS by using a hand refractometer, and 5 mL of supernatant was titrated by 0.05 NaOH with 1% phenolphthalein as an indicator.

### 2.5. Total Phenolic Content of Coconut Mesocarp

The total phenolic content was measured by the modified Folin–Ciocalteu reagent method. Two grams of coconut mesocarp were homogenized with 80% ethanol and then filtered using a sheet cloth. The crude extract was centrifuged at $12,000\times g$ at 4 °C for 20 min. The supernatant was diluted $50\times$ with double-ionized water. A total of 1 mL of the sample was mixed with 5 mL of 2 N Folin–Ciocalteu and then added to 4 mL of 7.5% sodium carbonate and incubated at 30 °C in the water bath for 30 min. Then, the test tube was held in cold water for 5 min, following which the absorbance (Abs) was measured at 760 nm by a spectrophotometer.

### 2.6. Polyphenol Oxidase (PPO) Activity of Coconut Mesocarp

A total of 1 g of coconut mesocarp was homogenized with 10 mL of a 0.1 M phosphate buffer (pH 7.0), to which 0.2 g of polyvinylpyrrolidone (PVP) was added. The homogenate was filtered with a sheet cloth. The homogenate was centrifuged at $17,400\times g$ and 4 °C for 20 min. The supernatants were collected. A total of 200 microliters (200 μL) of supernatant was added to 2750 μL of the 0.1 M phosphate buffer and 70 μL of 0.1 M catechol and then incubated at room temperature for 3 min. The absorbance was measured at 420 nm by a spectrophotometer. The protein analysis was performed using the Bradford assay [14]. The PPO activity was calculated as unit/mg protein.

### 2.7. pH of Coconut Mesocarp

After storage for one month, the coconut mesocarp was collected for pH measurement. A total of 5 g of coconut mesocarp were cut into small pieces and soaked in 50 mL of distilled water for 30 min, following which the pH was measured using a pH meter. The pH of the coconut mesocarp was indicative of the efficiency of $O_3UFBs$.

### 2.8. Microbiological Analysis

The total plate count was determined using a swab test on the surface of the trimmed coconuts with a sterile cotton swab. Serial dilutions were prepared in sterile 0.1% peptone solution (Merck, Darmstadt, Germany) and spread on duplicate plates of plate count agar (Merck, Germany) [15]. The plates were incubated for 48 h at $35 \pm 2$ °C. The microbial counts were expressed as CFU/g. Three coconut fruits from each treatment group were sampled after being kept at room temperature for three days.

### 2.9. Statistical Analysis

The experiments were repeated twice. The experiment had a completely randomized design (CRD). The data were recorded and analyzed by the SPSS version 16 software (IBM Corp., Armonk, NY, USA). Differences among the means were analyzed by a two-way analysis of variance (ANOVA), and the mean comparison was conducted using Duncan's new multiple-range test. Differences between the means at a 5% ($p < 0.05$) level were considered significant.

## 3. Results

### 3.1. Coconut Fruit Appearance

Trimmed coconuts stored at 2 °C for 0 days, and before storage (Figure 2A) for 30 days (Figure 2B) were then transferred to 8–10 °C for 7 days; no sign of the disease was found, and the mesocarp was still white, irrespective of the treatment group. The appearance of the fruit dipped in $C_{20}N_{10}$-$O_3UFBs$ and $C_{15}N_{15}$-$O_3UFBs$ when stored at 2 °C and simulated at on-shelf conditions for 7 days was the same as the appearance of the fruit treated using $C_{20}N_{20}$ (commercial method) (Figure 2).

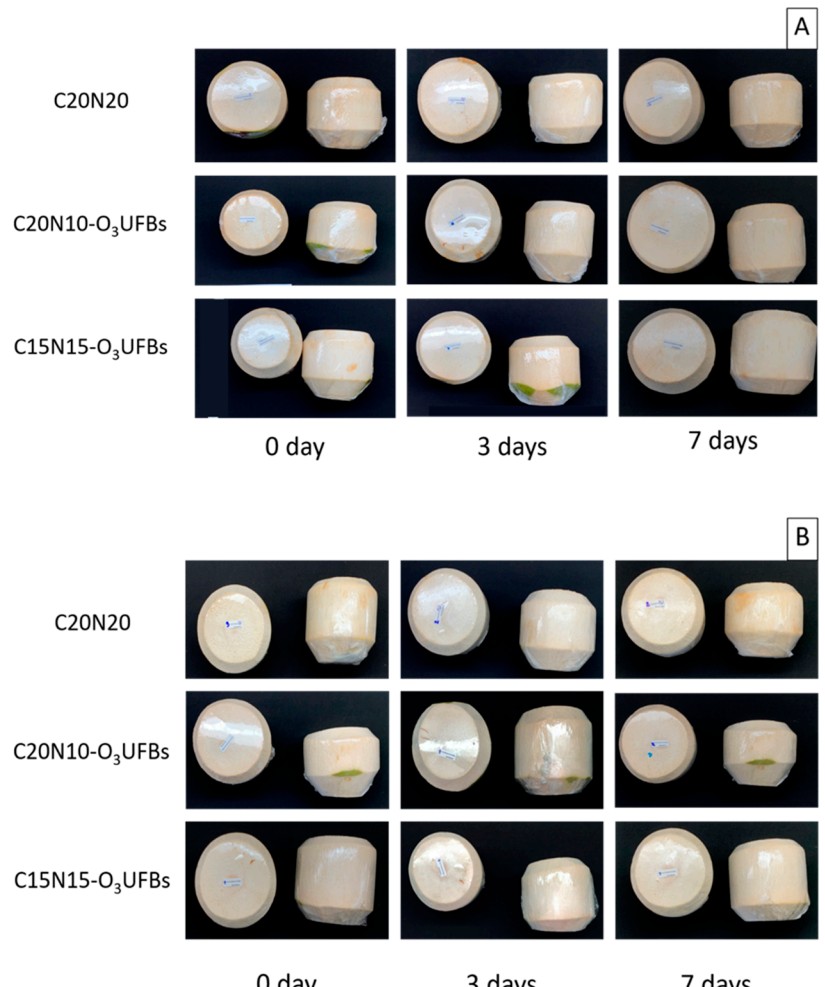

**Figure 2.** Trimmed coconuts dipped in $C_{20}N_{10}$-$O_3$UFBs and $C_{15}N_{15}$-$O_3$UFBs compared with $C_{20}N_{20}$ when stored at 2 °C for 0 day (before storage) (**A**) and 30 days (**B**). After storage, all coconut fruit were transferred to 8–10 °C for 7 days, and photos were taken on days 0, 3, and 7.

### 3.2. Disease Score

The disease score of the trimmed coconut before storage (Figure 3A) and after storage at 2 °C for 30 days (Figure 3B) were then transferred to 8–10 °C for 7 days and did not differ among the treatments. The score was about one point, which meant that no disease on the surface of the trimmed coconut was found (Figures 2 and 3).

### 3.3. Mesocarp Whiteness (W) and Browning Index (BI)

The mesocarp whiteness of trimmed coconut stored at 2 °C for 0 day (Figure 4A) was about 25–30, and the whiteness gradually increased after 30 days of storage (Figure 4B). The mesocarp whiteness of the coconut fruit dipped in $C_{20}N_{10}$-$O_3$UFBs, and $C_{15}N_{15}$-$O_3$UFBs did not differ from that dipped in $C_{20}N_{20}$ (Figure 4). This indicates that CA and NaCl combined with $O_3$UFBs had the same efficiency as the commercial method for preserving trimmed coconut.

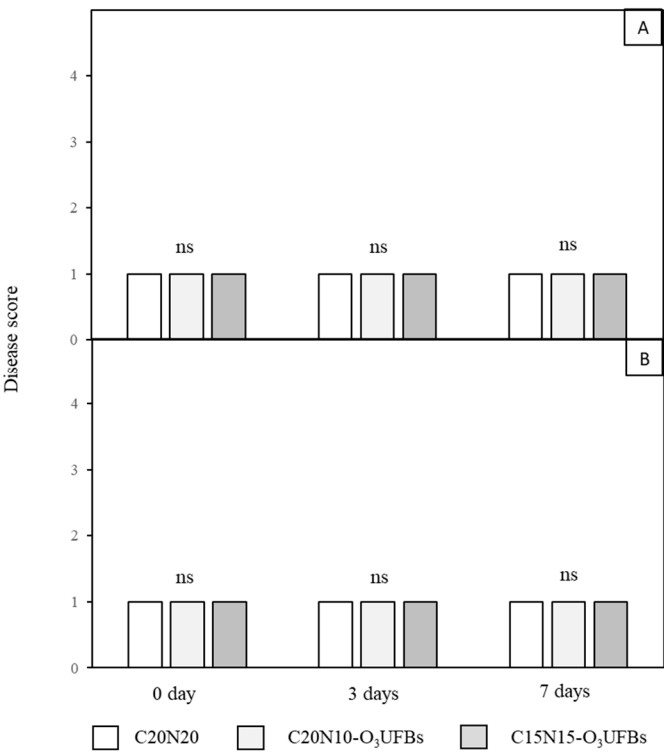

**Figure 3.** Disease scoring of trimmed coconuts dipped in $C_{20}N_{10}$-$O_3$UFBs and $C_{15}N_{15}$-$O_3$UFBs compared with $C_{20}N_{20}$ when stored at 2 °C for 0 days (before storage) (**A**) and 30 days (**B**). After storage, all coconut fruits were transferred to 8–10 °C for 7 days, and the fruit was evaluated for disease on days 0, 3 and 7. ns: not significantly different.

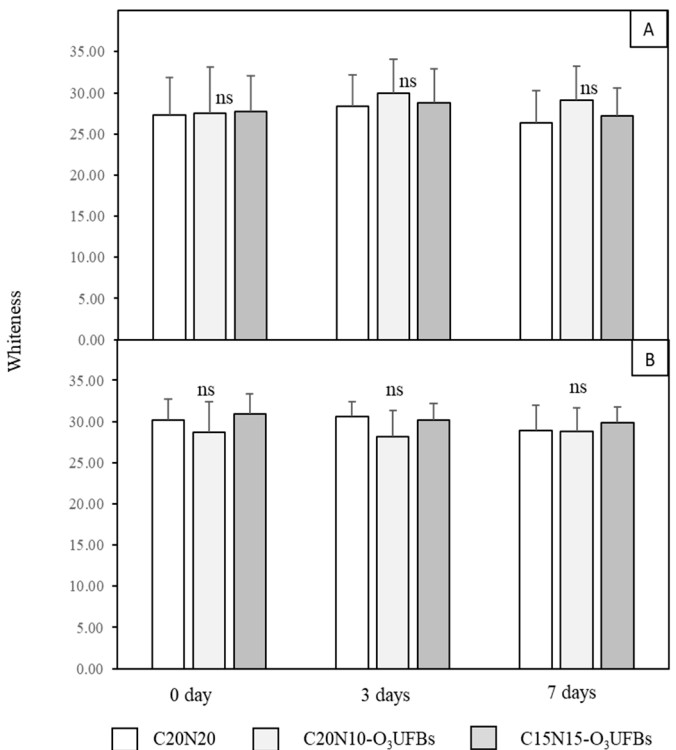

**Figure 4.** Mesocarp whiteness of trimmed coconuts dipped in $C_{20}N_{10}$-$O_3$UFBs and $C_{15}N_{15}$-$O_3$UFBs compared with $C_{20}N_{20}$ when stored at 2 °C for 0 days (before storage) (**A**) and 30 days (**B**). After storage, all coconut fruits were transferred to 8–10 °C for 7 days, and the fruits' whiteness was evaluated on days 0, 3, and 7. ns: not significantly different.

The browning index (BI) of coconut mesocarp was correlated with mesocarp whiteness (Figures 4 and 5). The BI did not differ significantly among the treatment groups when stored at 2 °C for 30 days and transferred to 8–10 °C for 7 days. The BI gradually decreased after a thirty-day storage period (Figure 5).

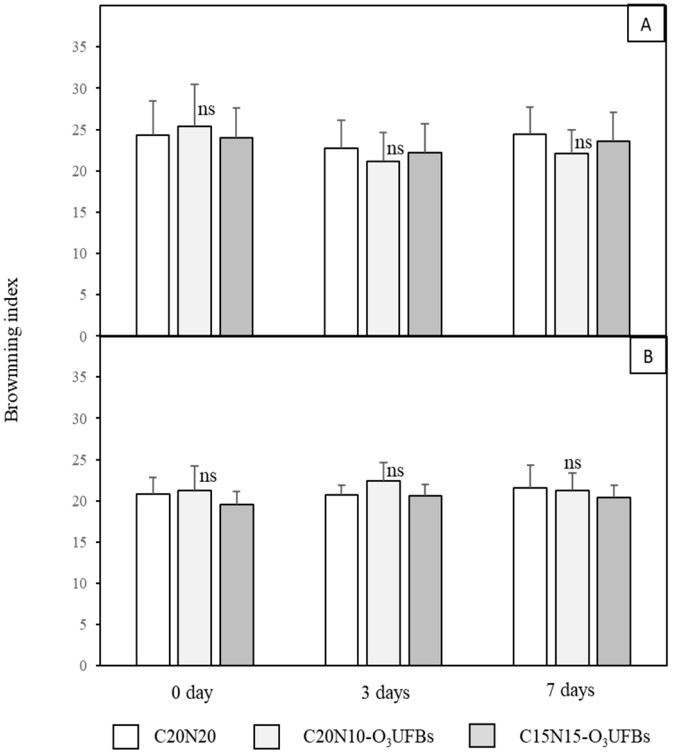

**Figure 5.** Mesocarp browning index of trimmed coconuts dipped in $C_{20}N_{10}$-$O_3$UFBs and $C_{15}N_{15}$-$O_3$UFBs compared with $C_{20}N_{20}$ when stored at 2 °C for 0 days (before storage) (**A**) and 30 days (**B**). After storage, all coconut fruits were transferred to 8–10 °C for 7 days, and the browning index was evaluated on days 0, 3, and 7. ns: not significantly different.

### 3.4. Coconut Water and Coconut Meat Qualities

#### 3.4.1. Coconut Water

The turbidity of coconut water had more than 85% transmittance across the treatment groups, which was clearly of the coconut water (Figure 6). The TSS of coconut water was around 7 °Brix when stored at 2 °C for 30 days, and it was not significantly different among the treatment groups after being transferred to 8–10 °C for 3 and 7 days (Figure 7A,B). The effect on the TA of coconut water across treatment solutions was the same as that of the commercial method. The TA of the coconut water was about 0.04% after being kept at 2 °C for 30 days, which was slightly decreased compared to before storage (0 days) (Figure 8A,B).

#### 3.4.2. Coconut Meat

The TSS of coconut meat was around 8–9 °Brix when stored at 2 °C for 30 days (Figure 7D). The TSS of coconut meat was higher than that of coconut water (Figure 7), and it was not significantly different among treatments after being transferred to 8–10 °C for 7 days (Figure 7C,D). The TA of the coconut meat of coconut dipped in $C_{20}N_{10}$-$O_3$UFBs, and $C_{15}N_{15}$-$O_3$UFBs did not differ from that treated with $C_{20}N_{20}$ (commercial method). The TA of coconut meat stored at 2 °C was the same after storage periods of 0 and 30 days (Figure 8C,D).

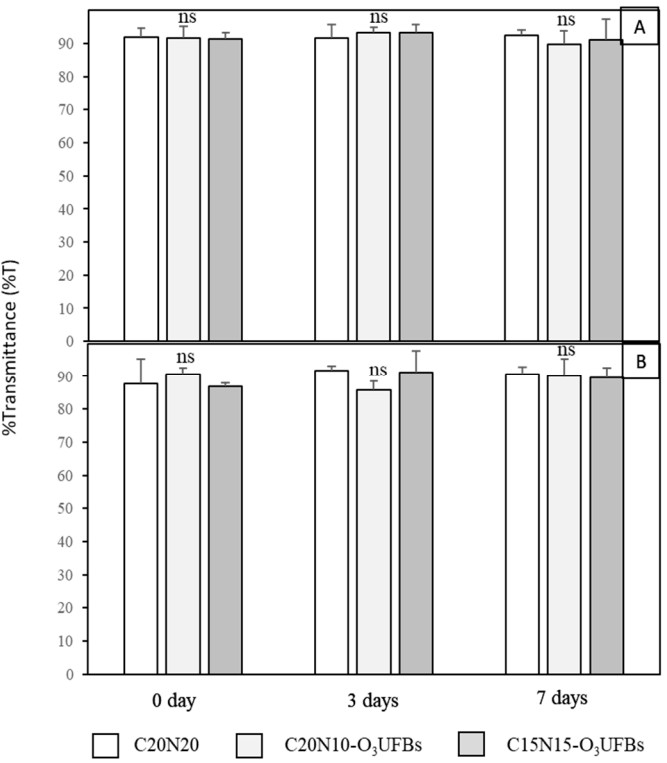

**Figure 6.** Turbidity of coconut water of trimmed coconuts dipped in $C_{20}N_{10}$-$O_3$UFBs and $C_{15}N_{15}$-$O_3$UFBs compared with $C_{20}N_{20}$ when stored at 2 °C for 0 days (before storage) (**A**) and 30 days (**B**). After storage, all coconut fruits were transferred to 8–10 °C for 7 days and coconut water turbidity was evaluated on days 0, 3, and 7. ns: not significantly different.

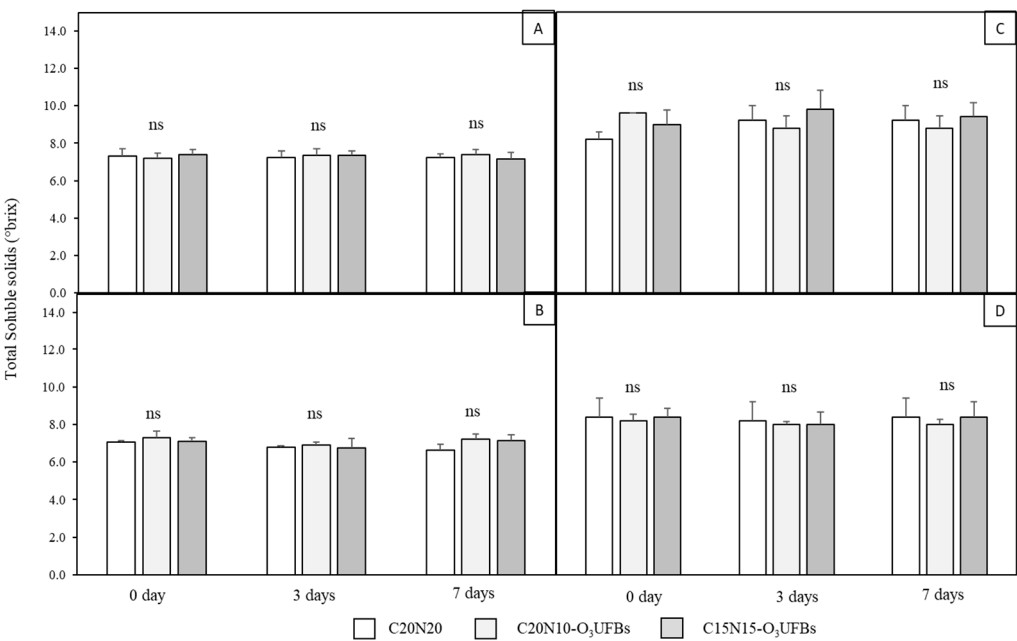

**Figure 7.** Total soluble solids (TSS) of coconut water (**A,B**) and coconut meat (**C,D**) for trimmed coconuts dipped in $C_{20}N_{10}$-$O_3$UFBs and $C_{15}N_{15}$-$O_3$UFBs compared with $C_{20}N_{20}$ when stored at 2 °C for 0 days (before storage) (**A**) and 30 days (**B**). After storage, all coconut fruits were transferred to 8–10 °C for 7 days and TSS were evaluated on days 0, 3, and 7. ns: not significantly different.

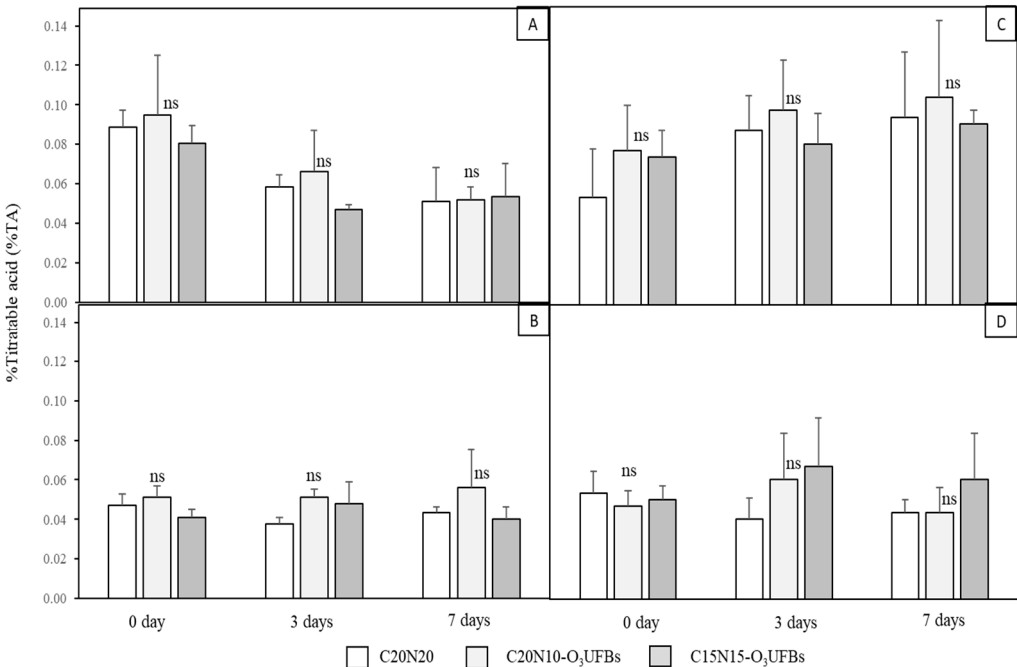

**Figure 8.** Titratable acidity (TA) of coconut water (**A**,**B**) and coconut meat (**C**,**D**) for trimmed coconuts dipped in $C_{20}N_{10}$-$O_3$UFBs and $C_{15}N_{15}$-$O_3$UFBs compared with $C_{20}N_{20}$ when stored at 2 °C for 0 days (before storage) (**A**) and 30 days (**B**). After storage, all coconut fruits were transferred to 8–10 °C for 7 days and TA was evaluated on days 0, 3, and 7. ns: not significantly different.

### 3.5. Total Phenolic Content of Coconut Mesocarp

The total phenolic content (TPC) of the coconut mesocarp (coconut husk) before storage at 2 °C and on-shelf simulation was not significantly different among the treatments (Figure 9A). After a thirty-day storage period, the TPC of the coconut husk did not differ significantly among the batches (Figure 9B). The TPC was about 4000–6000 mgGAE/gFW (Figure 9). The TPC of the trimmed coconut after storage for thirty days was transferred to 8–10 °C and appeared to be the lowest on day 7 (Figure 9).

### 3.6. PPO Activity of Coconut Mesocarp

The PPO activity of the coconut husk before storage at 2 °C and during on-shelf simulation was not significantly different among the treatments. On day 7, after being transferred to 8–10 °C, the PPO activity in the coconut husk was decreased compared to that recorded on days 0 and day 3 (Figure 10A). After being stored for 30 days, the PPO activity of the coconut husk across treatment batches was about 0.5–1.0 unit/mg protein and did not significantly differ among the treatments (Figure 10B). The PPO activity was concomitant with the TPC (Figures 9B and 10B).

### 3.7. pH of Coconut Mesocarp

The pH of the coconut mesocarp for trimmed coconut fruits dipped in $C_{20}N_{10}$-$O_3$UFBs, $C_{15}N_{15}$-$O_3$UFBs, and $C_{20}N_{20}$ after storage at 2 °C for 30 days was 4.18, 4.33, and 4.31, respectively. The pH of the coconut mesocarp for the trimmed coconut fruits dipped in $C_{20}N_{10}$ and $C_{15}N_{15}$ was 4.53 and 4.64, respectively, while the pH of the untreated coconut mesocarp was about 5.52.

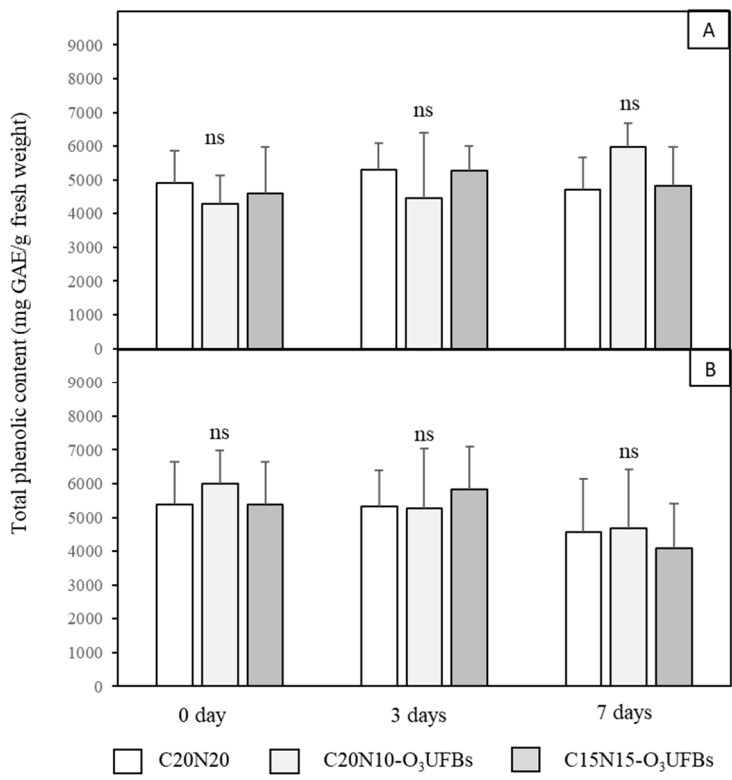

**Figure 9.** Total phenolic content of coconut mesocarp of trimmed coconuts dipped in $C_{20}N_{10}$-$O_3$UFBs and $C_{15}N_{15}$-$O_3$UFBs compared with $C_{20}N_{20}$ when stored at 2 °C for 0 days (before storage) (**A**) and 30 days (**B**). After storage, all coconut fruits were transferred to 8–10 °C for 7 days and total phenolic content was evaluated on days 0, 3, and 7. ns: not significantly different.

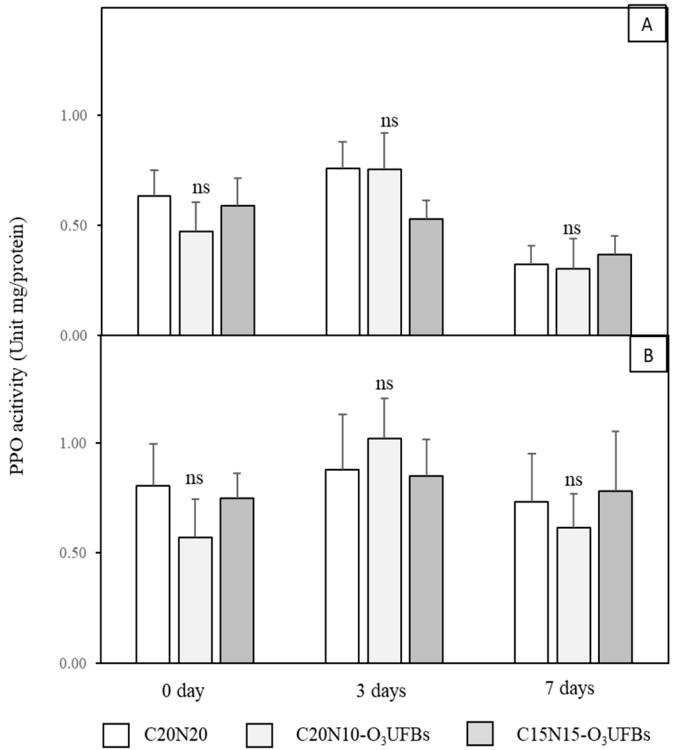

**Figure 10.** PPO activity of mesocarp of trimmed coconuts dipped in $C_{20}N_{10}$-$O_3$UFBs and $C_{15}N_{15}$-$O_3$UFBs compared with $C_{20}N_{20}$ when stored at 2 °C for 0 days (before storage) (**A**) and 30 days (**B**). After storage, all coconut fruits were transferred to 8–10 °C for 7 days and PPO was evaluated on days 0, 3, and 7. ns: not significantly different.

*3.8. Microbial Analysis (Preliminary Result)*

The efficacy of $O_3$UFBs treatment greatly affected the population of mesophilic bacteria on the trimmed organic aromatic coconut (Table 1). On day 3 at room temperature (RT), the total plate count of the coconuts dipped in $C_{20}N_{10}$-$O_3$UFBs, $C_{15}N_{15}$-$O_3$UFBs, and $C_{20}N_{20}$ was $5.0 \times 10^5$, $2.5 \times 10^6$, and $6.7 \times 10^5$ CFU/g, respectively (Table 1). There was no difference between the commercial method and $O_3$UFBs' treatment initially (Table 1).

**Table 1.** Total plate count and yeast and mold count of trimmed coconut dipped in $C_{20}N_{10}$-$O_3$UFBs or $C_{15}N_{15}$-$O_3$UFBs and compared with $C_{20}N_{20}$ after storage at room temperature for 3 days.

| Treatments | Total Plate Count (CFU/g) | Yeast and Mold (CFU/g) |
|---|---|---|
| $C_{20}N_{20}$ | $5.0 \times 10^5$ | $3.1 \times 10^8$ |
| $C_{20}N_{10}$-$O_3$UFBs | $2.5 \times 10^6$ | $8.2 \times 10^7$ |
| $C_{15}N_{15}$-$O_3$UFBs | $6.7 \times 10^5$ | $6.0 \times 10^7$ |
| F-test | ns | ns |

ns: not significantly different.

# 4. Discussion

The fruits' appearance, whiteness, and browning index (BI) after dipping in an acid/salt solution with $O_3$UFBs ($C_{20}N_{10}$-$O_3$UFBs and $C_{15}N_{15}$-$O_3$UFBs) were not significantly different from those treated using the commercial method. The whiteness and BI correlated with the total phenolic compound (TPC) and the PPO activity of coconut mesocarp. The browning of the trimmed young coconuts results from melanin generation due to the catalyzation of phenolic compounds by tyrosinase [16]. The pieces of coconut mesocarp without anti-browning agents ($O_3$UFBs water) showed surface browning, while a piece of coconut mesocarp and trimmed coconut fruit dipped in acid/salt with $O_3$UFBs displayed a white and bright appearance. CA and NaCl can reduce oxidation and pH, respectively, resulting in a reduction in PPO activity [17].

Interestingly, in all the treatments, the mesocarp whiteness increased after thirty days of storage, while the BI decreased compared with the previous storage (Supplementary Table S3). However, the decrease in BI correlated with TPC but did not correlate with PPO enzyme activity. In this study, the activity of PPO and TPC increased in the mesocarp of the trimmed coconut stored for one month. As a preliminary experimental result, the pieces of coconut husk dipped in CA only did not completely control the whiteness. Therefore, the result indicated that the increase in whiteness mainly resulted from the effect of NaCl, which is the chloride anion. In apples, the action was reported as a non-competitive reaction on inhibited PPO [18]. At low pH, $Cl^-$ and ionic strength might result in dramatic modifications of the enzyme conformation [19].

The effect of the acid/salt solution with $O_3$UFBs on coconut mesocarp after storage for 30 days did not differ from that of the commercial method. The pH of the coconut mesocarp after dipping in $C_{20}N_{10}$-$O_3$UFBs and $C_{15}N_{15}$-$O_3$UFBs was 4.18 and 4.33, respectively. However, the pH of coconut mesocarp after dipping in acid/salt solutions without $O_3$UFBs was about 4.53–4.64. $C_{20}N_{10}$-$O_3$UFBs resulted in a lower pH than the commercial method (pH: 4.31), but the difference was not significant. The mesocarp, when dipped in the acid/salt solution with $O_3$UFBs, showed the same level of enzyme activity as the commercial method. This indicates that the inhibitory effect of $O_3$UFBs on PPO activity was the same as the control treatment using $C_{20}N_{20}$. Our results are in agreement with those of Liao et al. [20], who reported the optimal pH of pear PPO activity to be 4.5–6.4.

The efficiency of the acid/salt solution with $O_3$UFB to control microbial growth was the same as $C_{20}N_{20}$ (commercial method). The treatment solution was able to control the growth of *Aspergillus* spp., *A. niger*, *Fusarium* spp., and *Penicillium* spp. on trimmed young coconuts under atmospheric conditions, as reported previously [21]. Low-pressure carbon dioxide microbubbles (MB-$CO_2$) controlled *Fusarium oxysporum* f. sp. melonis and *Pectobacterium carotovorum* in a hydroponic solution [22,23]. The generated UFBs with

ozone helped ozone dissolve well in the water and released a hydroxyl group (OH), thereby controlling microbial growth [24,25].

Nguyen et al. (2019) [7] reported that a 20% CA + 15% NaCl solution was the most effective treatment solution for maintaining the visual quality and color, as well as controlling microbial growth on trimmed young coconuts. However, the concentration of both chemicals that were needed to achieve this purpose is quite high. Treesuwan et al. [26] attempted to reduce the concentration of CA and NaCl by controlling the atmosphere conditions and offering a non-allergenic hurdle technology to prolong the shelf-life of trimmed young coconuts. However, controlled atmosphere technology cannot yet be used on a commercial scale in Thailand, whereas the ozone and ultrafine bubble technology was sustainable for commercial use. Our results show that dissolving acid/salt and ozone in ultrafine bubbles ($O_3$UFBs) is as efficient as the commercial method of preserving trimmed coconuts (20% CA + 20% NaCl). The combination of CA and NaCl at high concentrations inhibits enzymatic browning catalyzed by PPO and POD [27]. $O_3$UFBs can retain the efficiency of CA and NaCl even when used at reduced concentrations. Using $O_3$UFBs, the concentration of CA can be decreased from 20% to 15% and NaCl from 20% to 10%.

$O_3$UFBs with 20% CA + 10% NaCl and 15% CA + 15% NaCl had a low pH. Nano-ozone bubbles, or $O_3$UFBs, are capable of increasing the ozone concentration in water [8], and UFBs have a high surface area; these two factors help enhance the CA and NaCl efficiency and ability to remain suspended in water. The citric acid (CA) suppresses ozone self-decomposition in water due to hydroxy radicals ($HO^·$) [28]. The properties of ozone micro- and nano-bubbles (OMNBs) depend on the temperature, pH, salt concentration, and hydroxyl radicals. OMNBs ($O_3$UFBs) are stable under various salinity levels since they are negatively charged [24]. This results from the synergetic effect of $O_3$UFBs, CA, and NaCl. Acid/salt with $O_3$UFBs can control browning and microbial growth on trimmed young coconuts. Enzymatic browning is caused by the loose binding of copper at the active enzyme site of PPO; citric acid removes this copper [29], and $Cl^-$ anions replace the solvent ligand bridging the copper ions [30], thereby controlling the browning effect.

The TSS and TA of coconut water and coconut meat and the turbidity of coconut water did not differ significantly among the three treatment batches throughout the storage period, which is in agreement with the findings of Treesuwan et al. [26] and Luengwilai et al. [31]. The TSS content of coconut water was around 7 °Brix and slightly changed during storage, Supplementary Table S1, and on-shelf simulation. This indicated that TSS did not change much during storage, as reported previously [32]. During on-shelf simulation, the TA of coconut water tended to decrease, as shown in Supplementary Table S1, but the TA of coconut meat tended to be stable, as indicated in Supplementary Table S2. Interestingly, we also found that the content of TSS and TA for the coconut meat was higher than that of coconut water. The reduction in TSS and TA in coconut water, sugar, and organic acid in coconut water might play a role as a good source for the development of coconut meat. Because of the solid endosperm (coconut meat) developed after harvest, the coconut meat thickness increased during storage at 13 °C [32]. Moreover, the developing coconut meat absorbs the soluble materials in the coconut water, such as sugar [33], fatty acid [32], organic acid, and other materials.

## 5. Conclusions

Trimmed young coconuts treated with acid/salt-$O_3$UFBs solution ($C_{20}N_{10}$-$O_3$UFBs and $C_{15}N_{15}$-$O_3$UFBs) retained their qualities when stored for 30 days at 2 °C and for 7 days at 8–10 °C. During the storage period and on-shelf simulation, no significant difference was found among the coconuts treated in three different ways, in terms of TSS, TA, turbidity of coconut water, browning index, PPO activity, and total phenolic content.

Acid/salt-$O_3$UFBs reduced the concentration of citric acid by 25% and sodium chloride by 50% or 25% of the commercial concentration. Acid/salt-$O_3$UFBs is a promising new technique for controlling microbial growth and surface browning in trimmed young

coconuts. However, this technique must be tested on a commercial scale and undergo further investigation.

**Supplementary Materials:** The following supporting information can be downloaded at: https://www.mdpi.com/article/10.3390/horticulturae9020284/s1. Supplementary Table S1: Total soluble solids (TSS), titratable acidity (TA), and turbidity of coconut water of trimmed coconuts dipped in all chemical ($C_{20}N_{10}$-$O_3$UFBs, $C_{15}N_{15}$-$O_3$UFBs, and $C_{20}N_{20}$) before storage compared with after storage for 30 days. Supplementary Table S2: Total soluble solids (TSS) and titratable acidity (TA) of coconut meat of trimmed coconuts dipped in all chemical ($C_{20}N_{10}$-$O_3$UFBs, $C_{15}N_{15}$-$O_3$UFBs, and $C_{20}N_{20}$) before storage compared with after storage for 30 days. Supplementary Table S3: Whiteness, browning index, total phenolic content, and PPO activity of coconut mesocarp of trimmed coconuts dipped in all chemical ($C_{20}N_{10}$-$O_3$UFBs, $C_{15}N_{15}$-$O_3$UFBs, and $C_{20}N_{20}$) before storage compared with after storage for 30 days.

**Author Contributions:** Conceptualization, W.I. and S.J.; methodology, W.I., S.J. and S.P.; software, S.P.; validation, W.I. and S.P.; formal analysis, S.P. and W.I.; investigation, S.P.; resources, W.I. and S.J.; data curation, S.P. and W.I.; writing original draft preparation, S.P. and W.I.; writing—review and editing, W.I. and S.P.; visualization, S.P.; supervision, W.I.; project administration, W.I.; funding acquisition, W.I. and S.A. All authors have read and agreed to the published version of the manuscript.

**Funding:** This research was funded by the Kasetsart University Research and Development Institute (KURDI), grant number: FF(KU) 10.65.

**Data Availability Statement:** The data collected for this study are contained within the article.

**Acknowledgments:** We would like to thank K Fresh Co., Ltd., Samut Sakhon, Thailand, for providing us with trimmed aromatic coconuts for this study.

**Conflicts of Interest:** The authors declare no conflict of interest.

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
