# Peer review of "Ozone-Ultrafine Bubbles for Reducing Concentration of Citric Acid and Sodium Chloride for Trimmed Young Coconut Preservation"

_horticulturae, doi:10.3390/horticulturae9020284_

Round 1
Reviewer 1 Report
1) Section 2.3 must number the equations
2) Line 209-210 is a contradiction between the figure title and the image indices. the time varies from (0 days) that is how many hours or minutes, (3 days (72h)) then 7 days. (Figure 1. Trimmed coconuts dipped in C20N10-O3UFBs and C15N15-O3UFBs compared with C20N20 209 when stored at 2°C for 0 months (A) and 1 month (B) and then transferred to 10°C for 7 days.)
3) The same remark for all figures.
4) there are fewer references and also the references are very old
5) the results are misinterpreted
In general, the manuscript needs a thorough revision. For these reasons, I can accept this manuscript after a major revision.
Reviewer 2 Report
The manuscript investigated the feasibility of ozone-ultrafine bubble in the preservation of trimmed young coconut. The results showed that synergistic application of ozone-ultrafine bubble and citric acid/sodium chloride reduced the concentration of citric acid and sodium chloride. The research has application value. However, I have some concerns.
1. When the authors designed the experiment, why did the authors select the 20% CA + 10% NaCl + O3UFBs, 15% CA + 19 15% NaCl + O3UFBs? How about lower concentrations of CA and NaCl? How about only O3UFBs? Why did the authors not set a control (no any treatment)?
2. Please confirm the statistical analysis of Figure 8A.
Reviewer 3 Report
1. please check fig. 7. A and C. in A, the TA is gonig down from 0 day to 7 day. and in C TA is gong up from 0 days to 7 days.
2.there are no statistical analysis in Figure 2 and Table 1. the others figtures all overlape with line number, It is hard to know what those figtures meas.
Author Response
Please see the attachment.
The revised manuscript has been sent to the English editing service before resubmitting.

Round 2
Reviewer 1 Report
The same remark for all the manuscripts it is necessary to homogenize the unit of time.
0 months has no scientific meaning.
either you make all units in days or in hours.
Author Response
Please see in the attachment.
Reviewer 3 Report
This research is related to the preservation in young coconut. So the comparison between initiation stage and each storage point is very important and necessary. However, we cannot see any related information through statistical analysis in the manuscript, even though authors already investigated all the data what they need for it. The lack of these comparisons makes it impossible to present the value of this research, and it is difficult to achieve the novelty of your journal.
Author Response
Please see in the attachment.
